# Intake Estimation of Phytochemicals in a French Well-Balanced Diet

**DOI:** 10.3390/nu13103628

**Published:** 2021-10-16

**Authors:** Marie-Josèphe Amiot, Christian Latgé, Laurence Plumey, Sylvie Raynal

**Affiliations:** 1INRAE, MoISA, University of Montpellier, CIHEAM-IAMM, CIRAD, Institut Agro-Montpellier SupAgro, IRD, Campus La Gaillarde, 2 Place Pierre Viala, 34000 Montpellier, France; 2Pierre Fabre Laboratories, Langlade-3 Avenue Hubert Curien-BP 13 562, CEDEX 1, 31035 Toulouse, France; christian.latge@pierre-fabre.com; 3NUTRITION CO&CO, 11 Avenue des Vignes, 92210 St Cloud, France; laurence.plumey@free.fr; 4Naturactive, Pierre Fabre Laboratories, 29 Avenue du Sidobre, 81106 Castres, France; sylvie.raynal@pierre-fabre.com

**Keywords:** Mediterranean diet, phytonutrients, dietary recommendations, healthy diet, polyphenols, flavonoids, carotenoids, organosulfur, caffeine

## Abstract

Phytochemicals contribute to the health benefits of plant-rich diets, notably through their antioxidant and anti-inflammatory effects. However, recommended daily amounts of the main dietary phytochemicals remain undetermined. We aimed to estimate the amounts of phytochemicals in a well-balanced diet. A modelled diet was created, containing dietary reference intakes for adults in France. Two one-week menus (summer and winter) were devised to reflect typical intakes of plant-based foods. Existing databases were used to estimate daily phytochemical content for seven phytochemical families: phenolic acids, flavonoids (except anthocyanins), anthocyanins, tannins, organosulfur compounds, carotenoids, and caffeine. The summer and winter menus provided 1607 and 1441 mg/day, respectively, of total polyphenols (phenolic acids, flavonoids, anthocyanins, and tannins), the difference being driven by reduced anthocyanin intake in winter. Phenolic acids, flavonoids (including anthocyanins), and tannins accounted for approximately 50%, 25%, and 25% of total polyphenols, respectively. Dietary carotenoid and organosulfur compound content was estimated to be approximately 17 and 70 mg/day, respectively, in both seasons. Finally, both menus provided approximately 110 mg/day of caffeine, exclusively from tea and coffee. Our work supports ongoing efforts to define phytochemical insufficiency states that may occur in individuals with unbalanced diets and related disease risk factors.

## 1. Introduction

A healthy diet is considered to be one that provides adequate calories and nutrients to meet an individual’s needs for energy, growth, and repair, and to prevent diet-related non-communicable diseases. A diverse range of foods from several different food groups should be included. Plant-derived foods, such as whole grains, fruits, vegetables, nuts, and seeds, form an essential part of healthy diets.

Healthy, plant-rich diets (including Mediterranean-type or DASH (Dietary Approaches to Stop Hypertension) diets) have been reported to cover all of the macro- and micronutrients considered essential to health [1]. They are particularly rich in monounsaturated and polyunsaturated fatty acids and antioxidants, which regulate lipid and glucose metabolism, counteract oxidative stress, reduce inflammation, and support endothelial function [2]. Adherence to a Mediterranean diet has been associated with comparatively low rates of cardiovascular and cerebrovascular disease, diabetes, cancer, and cognitive decline [3,4,5,6,7,8].

Additionally, numerous studies have described the health benefits of foods rich in phytochemicals, also named bioactive compounds or phytonutrients or phytomicroconstituents, which are bioactive secondary metabolites found in plants, and in foods, drinks, and condiments derived from them [9,10]. Phytochemicals are chemically diverse compounds that can be classified into distinct families: polyphenols (phenolic acids, flavonoids (including anthocyanins), and tannins), carotenoids (for example, beta-carotene, lycopene, lutein, and zeaxanthin), organosulfur compounds (for example, isothiocyanates, indoles, allyl sulfur compounds, and sulforaphane), and phytosterols (for example, sitosterol, campesterol, stigmasterol, sitostanol, campestanol, and stigmastanol) [11]. Table 1 lists examples of foods that are high, medium, or low in each of these phytochemical families, although it must be noted that harvesting, processing, storage, and cooking can reduce the phytochemical content of food.

Dietary polyphenols are present in numerous plant-derived foods and beverages, their quantities differing according to the plant species, varietal, environmental factors (for example, climate and soil), and cultural practices. This family includes phenolic acids (for example, hydroxybenzoic and hydroxycinnamic acids), flavonoids, and tannins. Flavonoids, such as flavones, flavonols, flavanones, and flavanes (also called flavanols or catechins) are present in high amounts in dark chocolate (70–85% cocoa), black and green teas, herbs (for example, parsley, dill, shallots, fennel, mint, and thyme), salad vegetables (for example, red chicory, rocket, and cress), and grapefruit. Anthocyanins, a subtype of flavonoids, are water-soluble pigments that contribute to the red, purple, or blue color of plants and products derived from them. Thus, they are found in high quantities in berries (for example, blueberries, blackcurrants, blackberries, cranberries, and raspberries) and in red varieties of cabbage, chicory, and radish. Tannins are grouped into two main categories, namely hydrolyzable tannins (gallotannins and ellagitannins) and non-hydrolyzable, or condensed tannins (proanthocyanidins). Tannins are found in high quantities in berries (for example, blackcurrants, cranberries, blueberries, and strawberries), nuts (for example, hazelnuts, pecans, pistachios, almonds, and peanuts), stone fruits (for example, plums), red beans, and red cabbage.

Leafy greens and orange vegetables are the richest sources of carotenoids. The most abundant carotenoid is beta-carotene, which is hydrolyzed in vivo to vitamin A (hence its alternative name, provitamin A). However, other carotenoids, such as lycopene (a red pigment present in red tomatoes and grapefruit), lutein (found in green vegetables such as spinach and lettuce), and zeaxanthin (found in corn (maize)) are not metabolized to vitamin A.

Dietary sulfur compounds are divided into two classes: allyl sulfur compounds (for example, alliin), which are abundant in alliaceous vegetables, such as onions (*Allium cepa*) and garlic (*Allium sativum*); and glucosinolates, which are found in cruciferous vegetables, especially broccoli. During culinary processing and after ingestion, allyl sulfur compounds and glucosinolates undergo transformation to highly bioactive compounds (diallyl mono-, di- or tri-sulfides, sulforaphanes, and isothiocyanates) that have antioxidant and detoxifying properties [12].

A phytochemical index (PI), based on 24 h intake recall, has been proposed for use in epidemiological studies of diseases in which diet is a causative factor [13]. The PI is the percentage of dietary calories derived from foods that are rich in phytochemicals. Diets with a higher PI have been associated with a lower prevalence of cardiometabolic disorders, including abdominal obesity, hyperglycemia, hypertension, hypertriglyceridemia, and metabolic syndrome [13]. Although the PI is a useful global indicator, it does not consider the amounts of individual phytochemicals or phytochemical families that are present in a healthy diet; indeed, there is currently no sufficiently sophisticated estimation method that allows dietary phytochemical content to be readily quantified in a clinical setting. We therefore undertook a study to determine levels of different phytochemicals in a well-balanced French diet, in order to inform and support public health objectives and recommendations in France.

## 2. Materials and Methods

We designed two typical weekly seasonal menus, one for summer and one for winter, included plant-based foods commonly found in current French diets (see Appendix A). The menus were consistent with the current recommendations of the French National Agency for Food, Environmental and Occupational Health and Safety (ANSES; Agence Nationale de Sécurité Sanitaire Alimentation, Environnement, Travail) [14] and with the scientific literature [15]. Additionally, the menus were designed to include all essential macro- and micronutrients, and to account for seasonal variations in the availability of plant foods. We then prepared tables of the phytochemical content of each food, and used these to estimate the expected phytochemical intake per day for each menu.

Portion sizes were determined for each food category in accordance with French dietary habits, guidelines from the Groupe d’Étude des Marchés en Restauration Collective et Nutrition (GEM-RCN; French Collective Study Group on Catering and Nutrition) [16] and Programme National Nutrition Santé (PNNS; the French National Health Nutrition Programme) [17], and the opinion of the authors.

For each of the menus, the amount of each food category (FC) consumed daily (i.e., FCy, measured in grams) was calculated using the following formula:FC*y* = FF × *z* × PS
where FF (frequency factor) is a multiplier reflecting the frequency of intake (FF: Frequency factors = 1 per day, 0.143 per week, and 0.033 per month), z denotes the total number of servings, and PS is the portion size in grams.

A database of the phytochemical content of 116 plant foods (mainly fruits and vegetables) was created. The phytochemicals of interest were phenolic acids, flavonoids (including anthocyanins, which were considered separately); tannins; organosulfur compounds; carotenoids; and caffeine. The phytochemical content of individual foods was obtained from the US Department of Agriculture for carotenoids and flavonoids (including anthocyanidins) [18,19,20]; Phenol-Explorer for total polyphenols and phenolic acids [21,22,23]; and published scientific literature for tannins and organosulfur compounds [24]. Data for organosulfur compounds found in garlic and onion, regrouped in the same category, were also added. Other data were derived from consumption studies, such as the study of the average consumption of fruits and vegetables in France [14], as well as research on phytochemicals reported by Tennant et al. in 2014 [25].

Our analysis focused on phytochemicals with proven health benefits at amounts that can be readily obtained by dietary intake. Thus, we did not include phytosterols, because the beneficial effects of these compounds require a level of intake (for example, >3 g/day for phytosterols) that can only be achieved by dietary supplementation. Conversely, caffeine was included as a family in its own right because it accounts for a significant proportion of daily phytochemical intake, has well-documented health benefits, and is routinely included in nutritional recommendations such as those issued by ANSES [14].

Deviations of each phytochemical that result from an aggregation of numerous-level values associated to the variability of plant food products and their consumption frequencies are taken into account in our values. Moreover, some phytochemicals (i.e., carotenoids) supplied by consuming animal-origin foods such as fish, eggs, milk, and meats were not integrated in calculations. Animal food products were included in our menus to balance the caloric intake, but not retained, because their contribution was very low in comparison to that of the plant food products.

## 3. Results

For each menu, the weekly number of portions, and the estimated daily amounts of each food category, are shown in Table 2. The estimated daily amounts of phytochemicals associated with each menu are shown in Table 3.

### 3.1. Polyphenols

The summer and winter menus were estimated to provide 1607.4 and 1441.0 mg/day, respectively, of polyphenols. Approximately 50% of the polyphenol content of each menu was accounted for by phenolic acids (800 mg/day), with no difference being observed between menus. Among individual foods and food categories, filter coffee and starchy foods (including cereal grains such as oats, rye, and wheat) were the most important sources of phenolic acids, accounting for approximately 22–23% and 17–19%, respectively, of estimated total phenolic acids intake. Fruits and vegetables were also found to be important sources of phenolic acids, providing around 350 mg/day.

Flavonoids (including anthocyanins) and tannins were each estimated to account for approximately 25% of total polyphenols (400 mg/day in summer and 300–330 mg/day in winter; Table 3). Seasonal differences were driven by greater consumption of red and purple fruits and vegetables during the warmer months of the year, leading to a higher intake of anthocyanins in summer than in winter. Red and purple fruits and vegetables provided 287 mg/day of tannins in summer (71% of total daily tannin consumption), but just 113 mg/day in winter (37% of total daily tannin consumption). Conversely, pulses and white fruits and vegetables accounted for a greater proportion of total tannin intake in winter than in summer (46.7% vs. 18.4%, respectively).

### 3.2. Carotenoids

As shown in Table 3, the carotenoid content of each menu was significantly lower than the polyphenol content on a milligram basis (by a factor of 100 in summer and 84 in winter). However, there was little difference between the daily carotenoid content of the summer menu (16.2 mg/day) and the winter menu (17.1 mg/day) due to the year-round availability of foods such as leafy greens, orange fruits and vegetables, and tomatoes. Virtually all of the carotenoid content of both menus was provided by fruits and vegetables, with very little coming from other sources.

### 3.3. Organosulfur Compounds

Estimated intakes of organosulfur compounds were comparable in summer and winter (approximately 70 mg/day), but sources varied between the seasons. In summer, radishes formed an important source of organosulfur compounds, with cruciferous vegetables being important in the winter. *Allium* spp., such as garlic and onions, were important sources of organosulfur compounds all year round.

### 3.4. Caffeine

Our menus provided 111.6 mg/day of caffeine in both winter and summer, the majority of which was from coffee.

## 4. Discussion

We used public domain data to estimate the daily amounts of phytochemicals, divided into seven families, that would be provided by two one-week seasonal menus that were considered to: (i) represent a healthy diet; and (ii) accurately reflect the seasonal availability of plant foods in France.

The estimated total polyphenol content, calculated as the sum of phenolic acids, flavonoids, anthocyanins, and tannins of each menu was 1607.4 and 1441.0 mg/day in summer and winter, respectively. These values are 20–35% higher than those found in the French SUVIMAX study [26], but comparable to those in a recent analysis of data from the US National Health and Nutrition Examination Survey (NHANES) [27]. In the latter study, participants had mean total polyphenol intakes of 1656.6 mg/day, although there were significant differences between subgroups defined by age, sex, educational level, and body mass index.

We found coffee and starchy foods to be the most important sources of phenolic acids in our sample diets. Regular coffee drinkers have been shown to have a reduced risk of type 2 diabetes and cardiovascular diseases compared with non-drinkers [28]. Although the benefits of coffee have been assumed to be a result of its caffeine content [28], decaffeinated coffee also appears to provide health benefits [29]. This could be due to the presence of chlorogenic acids, which are not removed by decaffeination; these compounds are responsible for the antioxidant and anti-inflammatory properties of coffee [28]. Starchy foods also contain phenolic acids in significant amounts, with cereal bran being a particularly rich source of ferulic acid [30]. Epidemiological studies have suggested that diets rich in wholegrain cereals have protective effects against cardiovascular disease, type 2 diabetes, and cancer, and are useful in weight management; the latter observation may be attributable to both dietary fiber and phenolic acid content [31]. Consumption of leafy green vegetables, which we also found contributed significantly to phenolic acid intake, has been associated with a significantly reduced risk of type 2 diabetes [32]. This reduction, however, could be due to the presence of other phytochemicals (for example, carotenoids) and dietary fiber.

Our menus provided approximately 400 mg/day of flavonoids (including anthocyanins), a level of intake that was associated in a recent meta-analysis with the maximal reduction in coronary heart disease risk [33]. Among flavonoid subtypes, flavonols and flavones were associated with reduced risks of coronary heart disease, but anthocyanins and flavan-3-ols were found to offer the greatest protection against cardiovascular disease overall [33].

In vitro and animal studies have shown that flavonoids have antioxidant and anti-inflammatory properties, and that they can inhibit mutagenesis and carcinogenesis [34]. The potential of anthocyanins to prevent a range of chronic diseases (including obesity, type 2 diabetes, cardiovascular disease, visual disorders, and neuropathies) has also been studied [35]. For example, a meta-analysis of 128 randomized controlled trials reported that the consumption of foods rich in anthocyanins was associated with favorable changes in blood pressure and total cholesterol in overweight or obese people [36].

As well as being rich in anthocyanins, red and purple fruits and vegetables are important dietary sources of tannins. The reduced availability (and hence consumption) of these foods in winter contributes to an approximately 25% reduction in tannin intake during the colder months. This may be particularly relevant to the health of people who are overweight or obese, in whom the consumption of foods rich in ellagitannins (for example, pomegranate, berries, and nuts) has been shown to lower total cholesterol, low-density lipoprotein (LDL)-cholesterol, and triglycerides levels, and to reduce diastolic blood pressure [36]. With respect to procyanidins, (condensed tannins), Wallace and colleagues have proposed an intake of 200 mg/day, with an upper limit of 800 mg/day [37]. Tannins have been described as antinutritional factors because of their potential to impair iron bioavailability, and tannin consumption has been linked to the high prevalence of iron deficiency [38].

Overall, however, a diet rich in a diverse range of polyphenols may be recommended as an effective nutritional strategy to improve the health of patients with metabolic syndrome, with potentially beneficial effects on body fat, blood pressure, dyslipidemia, and insulin resistance, and reductions in oxidative stress, inflammation, and vascular dysfunction [39].

The antioxidant (singlet oxygen quenching and free radical scavenging) and anti-inflammatory properties of carotenoids are mediated via nuclear factor kappa B (NF-κB) receptors, which modulate cytokine and chemokine production [40]. Higher intakes of carotenoids, particularly beta-carotene and lycopene, have been shown to be inversely associated with a lower incidence risk of coronary heart disease and stroke [41]; furthermore, blood concentrations of carotenoids and cryptoxanthin are inversely associated with the incidence of cardiovascular disease, total cancer, and all-cause mortality [41]. Additionally, consumption of carotenoid-rich foods may reduce the risk of metabolic syndrome [42]. In the eye, lutein (present in leafy green vegetables), zeaxanthin, and meso-zeaxanthin (present in corn (maize)) accumulate in the macula; their beneficial effects on vision and eye health are due to protective effects against the oxidative damage caused by ultraviolet light [43]. Daily intakes of 6 mg of lutein and 18 mg of lycopene have been recommended, with upper limits of 60 and 50 mg/day, respectively [37].

Cruciferous vegetables are a source of glucosinolates, which are converted in vivo to sulforaphane; this bioactive metabolite has been found, in in vitro and animal studies, to have anticarcinogenic effects [44]. Allium plant species (for example, onion and garlic) are a source of S-alk(en)yl-L-cysteine sulfoxides. *Allium* spp. may provide some protection against cancer, cardiovascular disease, metabolic disorders, and bone disease, due to their antioxidant, anti-inflammatory, and lipid-lowering properties [45,46].

Our menus delivered identical daily amounts of caffeine regardless of season. In France, tea and coffee are the two major dietary sources of caffeine. At one end of the scale, brewed black tea provides approximately 20 mg caffeine per 100 mL, while, at the other, espresso coffee provides approximately 75 mg caffeine per 100 mL [47]. However, the caffeine content of prepared coffee beverages is known to vary considerably (58–259 mg per 100 mL), with espresso coffees typically containing less caffeine than brewed coffees [48], and there is therefore likely to be wide inter- and intra-individual variation in caffeine intake among French people. It is also important to recognize that, in recent decades, energy drinks and sodas have become significant sources of caffeine.

Caffeine, among other properties, has been reported to increase energy expenditure [49], and its consumption may have measurable metabolic effects in at least some individuals. In a meta-analysis published in 2011, caffeine and catechins were found to synergistically augment fat oxidation [50]. Rigorous reviews have concluded that the consumption of up to 400 mg/day of caffeine is not associated with adverse effects in healthy adults [51,52,53]. An upper limit of 300 mg/day has been proposed for pregnant women [51].

There are limitations to the type of analysis that we have presented here. First, we only considered 116 food items. Second, a major consideration is that agronomic practices (for example, variety, irrigation, fertilization, and harvesting date) and methods of food storage and preparation can significantly modify the phytochemical content of plant foods [54,55,56,57]. A sensitivity study should be carried out in the future with the high, medium, and low values of phytochemicals. We did not take this into account, due to a lack of robust scientific data in the USDA and Phenol-Explorer databases on these effects. The extent to which processing affects nutrient content varies between foods, and it was therefore impossible to apply a universal multiplier or other factor to adjust for food preparation. Indeed, while cooking food generally results in the loss of water-soluble, oxidizable, and heat-sensitive phytochemicals, it can also lead to an improvement in the assimilation of others, such as carotenoids [58].

## 5. Conclusions

Our study estimates the levels of different phytochemicals that are present in a well-balanced French diet (i.e., one meeting current guidelines on macro- and micronutrient requirements) in adults. This estimation constitutes the first step to addressing, in the future, adequate intakes to support health benefits. Our findings indicate that consuming a diverse range of plant foods provides a broad spectrum of phytochemicals in both summer and winter, with estimated intakes of each phytochemical family being generally comparable between the seasons. Further research linking dietary phytochemical intake to health outcomes would allow existing approaches to PI calculation to be refined and improved. We believe our work will help dietitians and nutritionists to identify gaps between observed and target phytochemical intake in adults in France, and to recommend personalized nutritional strategies for maintaining good health. We plan to carry out further work to deliver personalized dietary solutions, taking into account individual specificities, such allergies, and food intolerances and preferences.

## Figures and Tables

**Table 1 nutrients-13-03628-t001:** Foods and drinks considered to have high, medium, and low levels of different families of phytochemicals [11]. Reference ranges describing high, medium, and low phytochemical levels differ between phytochemical families, and are therefore shown alongside each family above the list of foods. The units of measurement for these reference ranges are mg per 100 g (for foods) or mg per 100 mL (for drinks).

Phytochemical Class	High	Medium	Low
Phenolic acids	100–650	45–100	5–45
	Flax and sunflower seeds, yams, red chicory, filter coffee, artichokes, prunes, mushrooms, endive, mangos, Jerusalem artichokes, raspberries	Cherries, chia seeds, dark chocolate (70–85% cocoa), pineapple, wholegrain wheat, flageolet beans, white beans, lentils, split peas	Watermelon, red beans, cashew nuts, walnuts, coriander (cilantro), potatoes, peaches, carrots, black tea, broccoli, dates, basil, apples, white rice, blackcurrants, quinces, green tea, nectarines, peaches, red wine, red bell peppers (capsicum), pears, strawberries, apricots, turnips, grapefruit, cauliflower
Flavonoids (except anthocyanins)	50–250	10–50	1–10
	Dark chocolate (70–85% cocoa), parsley, black tea, dill, shallots, fennel, green tea, red chicory, rocket, mint, grapefruit, cress, thyme	Red wine, blackberries, oranges, soy, kale, lemons, chia seeds, cranberries, onions, black grapes, artichokes, chives, asparagus, mandarins, blueberries, pecan nuts, buckwheat, cherries, olives, blackcurrants, turnips, pistachio nuts, broccoli, spinach, apricots, prunes, endive	Peaches, apples, almonds, figs, raspberries, green bell peppers (capsicum), Brussels sprouts, strawberries, bananas, nectarines, pears, coriander (cilantro), blackcurrants, hazelnuts, flat beans, flageolet beans, white beans, lentils, split peas, green beans, celery, garlic, leeks, white grapes, lettuce, kiwifruit, black radishes, chickpeas, quinces, persimmons, mangos, yellow bell peppers (capsicum), squash, potatoes, tomatoes, courgettes, flax seeds, cashew nuts, white cabbage, cauliflower, watermelon
Anthocyanins	45–210	10–45	1–10
	Red cabbage, blueberries, blackcurrants, red chicory, blackberries, aubergines (eggplants), cranberries, red radishes, raspberries	Currant berries, cherries, strawberries, red wine, pecan nuts	Pistachio nuts, hazelnuts, red beans, prunes, walnuts, almonds, apples, nectarines, pears, peaches, dates
Tannins	100–600	10–100	1–10
	Blackcurrants, red beans, hazelnuts, pecan nuts, cranberries, pistachios, red cabbage, plums, blueberries, almonds, peanuts, strawberries, apples	White and black grapes, peach, walnuts, currant berries, pears, apricots, raspberries, nectarines, red wine, blackberries, cherries, dates, mangos	Buckwheat, green tea, quinces, black tea, bananas, kiwifruit, cashew nuts
Carotenoids	5–20	2.5–5	1–2.5
	Spinach, carrots, parsley, kale, red chicory, basil, squash, yams, cress, lettuce, coriander (cilantro)	Arugula, watermelon, pistachio nuts, chives, leeks, thyme, tomatoes, persimmons, olives, peas	Red bell peppers (capsicum), grapefruit, apricots, melon, Brussels sprouts, broccoli, fennel, asparagus, flat beans, red cabbage, avocados
Sulfur compounds	>1000	100–250	10–100
	Onions, leeks	Brussels sprouts, garlic, black radishes, kale	Red radishes, red cabbage, broccoli, green cabbage, white cabbage, cauliflower
Caffeine	50–100	10–50	<10
	Espresso, filter coffee	Energy drinks, black tea, green tea	Cola drinks
Phytosterols	>200	50–200	3–50
	Vegetable oils (corn, sunflower, soybean)	Almonds, peanuts, corn (maize), oats, wheat	Cauliflower, broccoli, carrots, tomatoes, apples, bananas, grapes, oranges

**Table 2 nutrients-13-03628-t002:** Menu analysis, according to the number of portions of each food category consumed per week and the amount (expressed in grams) of each food category consumed per day.

Food Category	Examples	Portions per Week		Amounts per Day (g)	
		Summer Menu	Winter Menu	Summer Menu	Winter Menu
Grains, beans, nuts, and seeds					
Starchy foods	Corn, buckwheat, Jerusalem artichokes, oats, potatoes, quinoa, rice (white and whole), rye, wheat (refined and whole)	16	18	274.3	308.6
Pulses	Chickpeas, flageolet beans, lentils, red beans, split peas, white beans	2	2	34.3	34.3
Nuts and seeds	Almonds, cashew nuts, chia seeds, coconut, flax seeds, hazelnuts, macadamia nuts, peanuts, pecan nuts, pistachio nuts, sesame seeds, sunflower seeds, walnuts	0	1	0	2.9
Fruits, vegetables, spices, and herbs					
White	Apples, endive, mushrooms, quinces, parsnips, pears, salsify, turnips, white grapes	2	13	35.7	232.1
White/green	Celery, fennel, leek, shallots	2	3	35.7	53.6
Green	Avocados, asparagus, artichokes, courgettes, cress, cucumbers, flat beans, green beans, green bell peppers (capsicum), kiwifruit, lettuce, olives, peas, rocket, spinach	10	7.5	178.6	133.9
Yellow	Bananas, lemons, pineapple, yellow bell peppers (capsicum)	3	5	53.6	89.3
Orange	Carrots, mandarins, oranges, squash, yams	10	6	178.6	107.2
Red	Beetroot, cherries, cranberries, redcurrants, dates, figs, grapefruit, plums, raspberries, red bell peppers (capsicum), strawberries, tomatoes, watermelon	13	2	232.1	35.7
Purple	Plums, blackcurrants, blackberries, blueberries, eggplants, black grapes, red chicory	5	3	89.3	53.6
Cruciferous vegetables	Brussels sprouts, broccoli, cauliflower, green cabbage, kale, red cabbage, white cabbage	0	2	0	35.0
Radishes	Black radishes, red radishes	2	0	35.0	0
*Allium* spp.	Garlic, onions, chives	2	2	5.6	5.6
Herbs	Basil, coriander (cilantro), dill, mint, parsley, thyme	10	9	14.3	12.9
Beverages					
Coffee (filter)		3	3	85.7	85.7
Coffee (espresso)		7	7	60.0	60.0
Tea	Black tea, green tea	4	4	114.3	114.3
Wine	Red wine	2	2	34.3	34.3
Other	Sodas, energy drinks	1	1	28.0	28.0
Plant-derived sweets					
Chocolate	Dark chocolate (70–85% cocoa)	7	7	19.6	19.6

**Table 3 nutrients-13-03628-t003:** Quantity of each phytochemical family (expressed in mg/day) provided by the menus, according to food category.

Food or Beverage Category	Phytochemical Content (Summer) [mg/Day]							Phytochemical Content (Winter) [mg/Day]						
	PA	F	A	T	Car	OS	Caf	PA	F	A	T	Car	OS	Caf
Grains, beans, nuts, and seeds														
Starchy foods	136.0	4.2	0.0	2.5	0.2	0.0	0.0	153.0	4.7	0.0	2.8	0.3	0.0	0.0
Pulses	14.8	0.8	0.8	61.9	0.0	0.0	0.0	14.8	0.8	0.8	61.9	0.0	0.0	0.0
Nuts and seeds	0.0	0.0	0.0	0.0	0.0	0.0	0.0	5.3	0.2	0.1	4.9	0.0	0.0	0.0
Fruits, vegetables, spices, and herbs														
White	21.0	1.7	0.2	12.2	0.0	0.0	0.0	136.6	11.2	1.1	79.0	0.1	0.0	0.0
White/green	0.2	18.2	0.0	0.0	0.4	0.0	0.0	0.3	27.3	0.0	0.0	0.6	0.0	0.0
Green	106.3	25.3	0.0	0.4	5.9	0.0	0.0	79.7	18.9	0.0	0.3	4.5	0.0	0.0
Yellow	17.2	1.5	0.0	0.5	0.0	0.0	0.0	5.7	22.8	0.0	0.2	0.3	0.0	0.0
Orange	98.1	6.7	0.8	25.7	5.3	0.0	0.0	100.0	1.5	0.2	3.2	8.9	0.0	0.0
Red	94.2	24.8	38.8	132.4	2.3	0.0	0.0	14.5	3.8	6.0	20.4	0.4	0.0	0.0
Purple	46.2	25.4	96.5	154.2	1.4	0.0	0.0	27.7	15.3	57.9	92.5	0.8	0.0	0.0
Cruciferous vegetables	0.0	0.0	0.0	0.0	0.0	0.0	0.0	2.9	2.8	10.5	24.5	0.7	28.7	0.0
Radishes	0.5	0.0	0.0	11.0	0.0	30.7	0.0	0.0	0.0	0.0	0.0	0.0	0.0	0.0
*Allium* spp.	0.1	0.8	0.0	0.0	0.0	39.0	0.0	0.1	0.8	0.0	0.0	0.0	39.0	0.0
Herbs	4.2	9.9	0.0	0.0	0.6	0.0	0.0	3.8	8.9	0.0	0.0	0.6	0.0	0.0
Beverages														
Coffee (filter)	181.8	0.0	0.0	0.0	0.0	0.0	44.0	181.8	0.0	0.0	0.0	0.0	0.0	44.0
Coffee (espresso)	60.8	0.0	0.0	0.0	0.0	0.0	42.8	60.8	0.0	0.0	0.0	0.0	0.0	42.8
Tea	17.9	110.6	0.0	4.5	0.0	0.0	18.3	17.9	110.6	0.0	4.5	0.0	0.0	18.3
Wine	4.2	16.6	7.5	7.7	0.0	0.0	0.0	4.2	16.6	7.5	7.7	0.0	0.0	0.0
Soda	0.0	0.0	0.0	0.0	0.0	0.0	6.6	0.0	0.0	0.0	0.0	0.0	0.0	6.6
Plant-derived sweets														
Chocolate (dark, 70–85% cocoa)	15.9	51.4	0.0	0.0	0.0	0.0	0.0	15.9	51.4	0.0	0.0	0.0	0.0	0.0
Total (mg)	802.8	247.0	155.7	401.9	16.4	69.8	111.6	809.0	246.2	83.9	301.9	17.1	67.7	111.6
Total polyphenols (PA + F + A + T)	1607.4							1441.0						

A, anthocyanins; Caf, caffeine; Car, carotenoids; F, flavonoids (except anthocyanins); OS, organosulfur compounds; PA, phenolic acids; T, tannins.

## Data Availability

All scientific data calculated in the course of this study, and presented in this publication, are available from Christian Latgé and Sylvie Raynal upon reasonable request. All requests should be sent to: sylvie.raynal@pierre-fabre.com.

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
