# Peer review of "Intake Estimation of Phytochemicals in a French Well-Balanced Diet"

_nutrients, 2021, doi:10.3390/nu13103628_

Round 1

Reviewer 1 Report

In this paper, authors have made serious effort to emphasise the role of phytonutrients towards the health benefits of plant-rich diets.

They targeted their study to estimate the amounts of phytonutrients in a well-balanced diet. The work focussed mainly on phenolic acids, flavonoids (except anthocyanins), anthocyanins, tannins, organosulfur compounds, carotenoids, and caffeine. To different types of menus were used in the study.

Finally, authors concluded about the phytonutrient insufficiency states that may occur in individuals with unbalanced diets and related disease risk factors in France.

Authors hope that their work help dietitians and nutritionists to identify gaps between observed and target phytonutrient intake in adults in France. This work appears to have great significance for dietitians and nutritionists of the country for directing their adults to adapt to a healthy diet containing phytonutrients and avoid diseases.

Major comments

  1. Authors have considered two different menus (winter and summer), But how did they ensure the intake of only these phytonutrients without any deviation.
  2.  Authors have suggested a proper diet containing phytonutrients, ideally, it is accepted, but in practice people also consume meat based foods. Practically, people consume whole lot of varieties of other foods. In real scenario, what will be the limitation of this study?
  3.  Authors did not talk about an individual’s allergy towards a specific phytonutrient based diet.

Author Response

1.  Authors have considered two different menus (winter and summer), But how did they ensure the intake of only these phytonutrients without any deviation.

Thank you for this comment. Deviations of each phytonutrient are taken into account in the values ​​that are extracted from composition tables and resulted from an aggregation of numerous level values associated to the variability of food products and their consumption frequencies.  To fully respond to your request, a sensitivity study should be carried out with the high, medium and low values. We added a comment in methods  (Deviations of each phytochemical are taken into account in our values that result from an aggregation of numerous level values associated to the variability of plant food products and their consumption frequencies) and in the discussion, more specifically, in the "limitations" paragraph (A sensitivity study should be carried out in the future with the high, medium and low values of phytochemicals)

2. Authors have suggested a proper diet containing phytonutrients, ideally, it is accepted, but in practice people also consume meat based foods. Practically, people consume whole lot of varieties of other foods. In real scenario, what will be the limitation of this study?

We assume that some phytochemicals (carotenoids) can be supplied by consuming animal-origin foods such as fish, eggs, milk, meats. We have included these food product in our menus to balance the caloric intake but not retained in our calculation for the different phytochemical families because their contribution remains very low versus the plant ones.

We added in Material and methods: Moreover, some phytochemicals (i.e carotenoids) supplied by consuming animal-origin foods such as fish, eggs, milk and meats were not integrated in calculations. Animal food products were included in our menus to balance the caloric intake but not retained because their contribution was very low versus the plant ones.

3. Authors did not talk about an individual’s allergy towards a specific phytonutrient based diet.

It is a relevant remark, but at this first stage we did not take specific criteria, such allergy,  in our menus that have been targeted the general population. In our work, we assume that allergy to a single food has a low impact on phytochemical levels. If there is a specific allergy, we can propose substitution in menus to reach the intake of each phytochemical family. We plan to carry further work to deliver personalized dietary solutions, not only for allergic people but  also for food intolerances, preferences ...

We added in the conclusion the following sentence: We plan to carry further work to deliver personalized dietary solutions, taking into account individual specificities, such allergies, food intolerances and preferences.

Reviewer 2 Report

This is very comprehensive information on the real intake of Phytonutrients, and it would be very helpful to include a table with the quantities associated to certain benefits based on the studies you mentioned. In this case, your paper would clarify how close or far are to doses that have a health impact.

Could you include it?

In the attachments, there are misspellings that need to be adjusted.

Author Response

1. This is very comprehensive information on the real intake of Phytonutrients, and it would be very helpful to include a table with the quantities associated to certain benefits based on the studies you mentioned. In this case, your paper would clarify how close or far are to doses that have a health impact.

Could you include it?

 Regarding your justified request, we plan to submit another manuscript in which we are currently to describe the health benefits of phytochemicals, the diverse health targets and  their dose effects  when reported in literature. 

In the present work, we focused on the upper doses of some phytochemicals (tannins, caffeine and carotenoids) to check if our menus did not induce health risks. For other phytochemicals, the risk is limited due to their low bioavailability (low intestinal absorption and high metabolization).

We guess that a new table is not necessary because one has been established by Wallace et al (cited in our manuscript) who reported specific proposed levels and upper limits for the same phytochemicals.

2. In the attachments, there are misspellings that need to be adjusted.

We did not see what are misspellings (no attachment)